# Reconfiguration of functional brain networks and metabolic cost converge during task performance

**Andreas Hahn[1]\*, Michael Breakspear[2,3], Lucas Rischka[1], Wolfgang Wadsak[4,5], Godber M Godbersen[1], Verena Pichler[4], Paul Michenthaler[1], Thomas Vanicek[1], Marcus Hacker[4], Siegfried Kasper[1], Rupert Lanzenberger[1], Luca Cocchi[2]**

[1]Department of Psychiatry and Psychotherapy, Medical University of Vienna, Vienna, Austria; [2]QIMR Berghofer Medical Research Institute, Brisbane, Australia; [3]School of Psychology, University of Newcastle, Newcastle, Australia; [4]Department of Biomedical Imaging and Image-guided Therapy, Division of Nuclear Medicine, Medical University of Vienna, Vienna, Austria; [5]Center for Biomarker Research in Medicine (CBmed), Graz, Austria

**Abstract** The ability to solve cognitive tasks depends upon adaptive changes in the organization of whole-brain functional networks. However, the link between task-induced network reconfigurations and their underlying energy demands is poorly understood. We address this by multimodal network analyses integrating functional and molecular neuroimaging acquired concurrently during a complex cognitive task. Task engagement elicited a marked increase in the association between glucose consumption and functional brain network reorganization. This convergence between metabolic and neural processes was specific to feedforward connections linking the visual and dorsal attention networks, in accordance with task requirements of visuo-spatial reasoning. Further increases in cognitive load above initial task engagement did not affect the relationship between metabolism and network reorganization but only modulated existing interactions. Our findings show how the upregulation of key computational mechanisms to support cognitive performance unveils the complex, interdependent changes in neural metabolism and neuro-vascular responses.

\*For correspondence:
andreas.hahn@meduniwien.ac.at

## Introduction

Brain function relies on coordinated activity in local neural circuits, from which large-scale functional brain networks are composed (*Park and Friston, 2013*). Such changes in the activity of local neural circuits and large-scale systems are paralleled by the modulation of metabolic processes (*Riedl et al., 2016*). Recent work has assessed the dynamic changes in functional brain network architecture associated with cognitive task engagement. Results from these investigations have challenged the prior notion (*Smith et al., 2009*) that functional interactions at resting-state are relatively stable and sufficient to support goal-directed behavior (*Cocchi et al., 2013*). For example, studies have highlighted dynamic brain network reconfigurations when switching from a state of rest to that of cognitive engagement (*Braun et al., 2015*). Solving tasks of increasing cognitive complexity has further been linked to complex changes in the functional interplay between cortical regions that otherwise comprise distinct networks at rest (*Hearne et al., 2017*). These findings are in line with the proposal that brain networks exhibit a flexible modular architecture, with highly interconnected hub regions changing their functional network assignment according to task demands (*Cole et al., 2013*).

Task-induced neural activation in segregated brain regions and corresponding network dynamics have, of late, been largely assessed using the blood oxygen level dependent (BOLD) signal. The BOLD signal represents a non-specific proxy of activation, which is directly mediated by hemodynamic factors including changes in blood flow and oxygen content rather than by neural metabolism (*Heeger and Ress, 2002*). As a result, the metabolic underpinnings accompanying large-scale functional network reconfigurations that support cognition have received little recent research focus (*Horwitz et al., 1995*; *McIntosh et al., 1994*). It also remains unknown if the association between functional connectivity and metabolism is confined to specific task-related brain networks or extends more globally. Furthermore, the variation of this association as a function of task load is poorly understood. To address this gap in knowledge, we analyzed simultaneously acquired positron emission tomography (PET) and functional magnetic resonance imaging (fMRI, i.e., BOLD and arterial spin labeling (ASL)) data while healthy participants performed a well-validated cognitive task comprising two levels of difficulty (*Haier et al., 2009*). By combining these different imaging modalities we were able to (i) comprehensively map whole-brain network reconfigurations as a function of varying cognitive demands; (ii) assess the coupling between task-driven metabolic increases and modulations in functional connectivity; and (iii) model the neural dynamics underpinning changes in functional connectivity and related metabolic demands. We hypothesized significant task-induced changes in the link between metabolic and neural factors, supporting our ability to flexibly engage in cognitive processes. Previous work has shown that the bulk of functional network reconfigurations occur when participants engage in a cognitive task, with minor reorganizations as a function of cognitive load (*Hearne et al., 2017*). We therefore expect greater metabolic demands to be mobilized to establish task-specific patterns of functional brain interactions.

## Results

Simultaneous functional MRI (BOLD and ASL) and [$^{18}$F]FDG PET data were acquired while 22 healthy adult participants performed a cognitive task (Tetris). The task had two different levels of difficulty and involved rapid visuo-spatial processing and motor coordination (see Materials and methods and *Figure 1A and B*).

We first integrated information regarding task-related changes in local BOLD signals, cerebral blood flow (CBF) and glucose consumption to identify functionally segregated brain regions involved in task performance (*Figure 1C*). This analysis leveraged the recently introduced approach of functional PET (fPET) imaging to determine task-specific changes in the cerebral metabolic rate of glucose (CMRGlu) (*Hahn et al., 2016*; *Rischka et al., 2018*) as well as canonical analyses for BOLD and ASL signal changes. We then assessed task-based network reconfigurations between the ensuing identified brain regions using metabolic connectivity mapping (MCM) (*Riedl et al., 2016*; *Figure 1C*). MCM evaluates the association between *regional* patterns of BOLD-derived functional connectivity and glucose metabolism (see Materials and methods for rationale underlying this approach). To this end, task-based functional connectivity was computed from data collected during periods of continuous task performance (6 min, *Figure 1A*). This approach minimizes the influence of nonspecific factors including episodic rest-to-task transitions and major changes in visual luminance (*Cole et al., 2019*). We finally tested the link between metabolic and neural dynamics by comparing results from dynamic causal modeling (DCM) to those obtained using MCM (*Figure 1C*).

### Behavioral results

Cognitive performance differed significantly between the two task levels (*Figure 1B*). As expected, fewer lines were completed (easy 40.5 ± 12.5 *vs.* hard 32.5 ± 16.7, p = 0.015) and more games lost in the hard task trials compared to the easier ones (easy 0.5 ± 1.0 *vs.* hard 9.8 ± 3.4, p = $10^{-11}$).

### Changes in local metabolism and neural activity as a function of task difficulty

Relative to resting-state, task performance elicited significant increases in CMRGlu, CBF and BOLD-signals (all $p_{FWE} < 0.05$, *Figure 2*). These cross-modal changes showed a high spatial overlap among each other (*Figure 2*, Dice coefficient = 0.49–0.55). Significant changes across the three modalities overlapped in the occipital cortex (Occ), the supplementary motor area (SMA), the intraparietal sulcus (IPS) and the frontal eye field (FEF, *Figure 2* conjunction map). Increases in task difficulty (hard

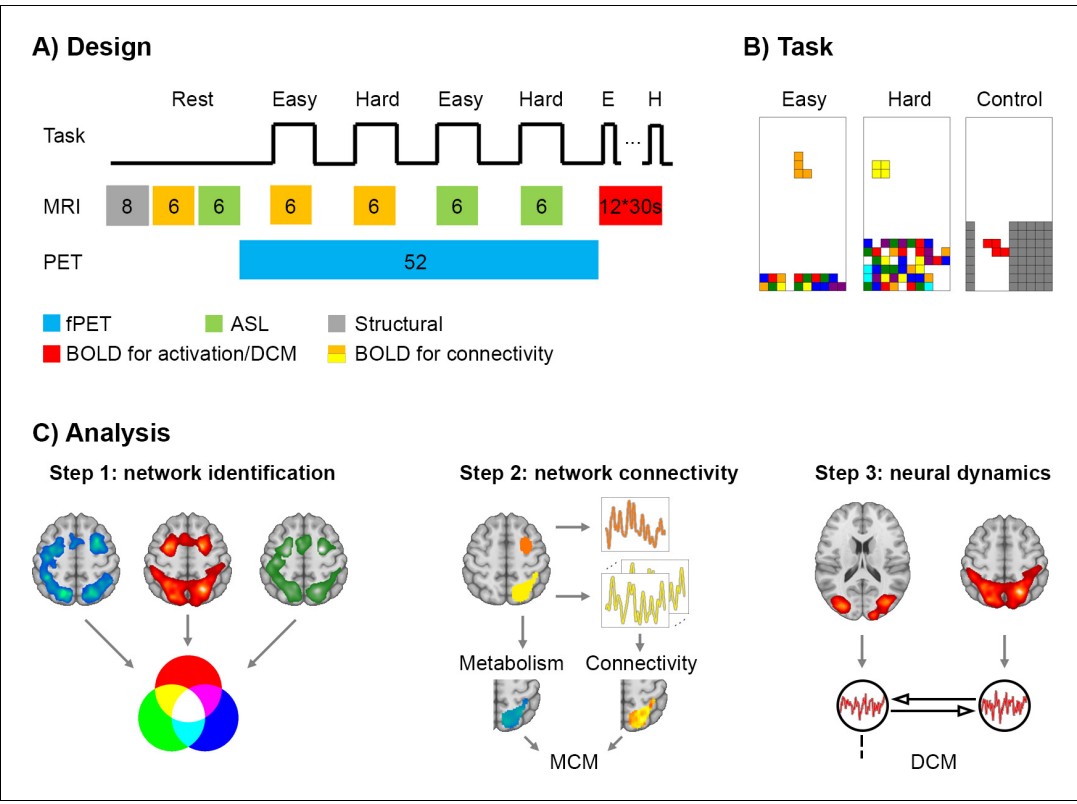

**Figure 1.** Design and work flow. (**A**) The experimental sequence comprised a T1-weighted structural scan (8 min, grey), BOLD (6 min, orange) and ASL (6 min, green) at rest. This was followed by task-specific PET/MR acquisition. During the fPET measurement (52 min, blue) participants completed four times (6 min each) the video game Tetris with two levels of cognitive load. Simultaneous MRI acquisition included BOLD used to estimate functional connectivity (orange) and ASL (green). In the final part of the experiment, participants performed 12 task blocks (easy, hard and control, 30 s for each block, red) during BOLD acquisition to estimate neural task effects and to compute effective connectivity using dynamic causal modeling. Numbers indicate the duration of blocks in minutes unless indicated otherwise. (**B**) Task load (easy and hard) was defined by the speed of the falling bricks and the amount of bricks at the bottom. In the control condition the bricks were navigated through the channel, but no lines could be built as bricks vanished afterwards. (**C**) Data analysis comprised three main steps (same color codes as in A). First, brain regions involved in task processing were identified as conjunction of task-specific changes in glucose metabolism (fPET), blood oxygenation (BOLD) and blood flow (ASL). Second, the interplay between these brain regions was determined by the combination of metabolism and functional connectivity at rest and during task performance. Seed-to-voxel correlations were calculated to obtain patterns of functional connectivity. Metabolic connectivity mapping (MCM) (*Riedl et al., 2016*) was then computed by correlating the regional patterns of metabolic demands and functional connectivity with the inference of directionality if these spatial patterns show a significant association. Third, to assess the putative link between MCM and neural dynamics, the resulting MCM model was compared to the one obtained using dynamic causal modeling (DCM).

*vs.* easy) caused a significant increase in the imaging parameters in these brain regions (all $p_{Bonf\text{-}Holm}$ = $10^{-7}$ to 0.020), with the exception of CMRGlu in SMA as well as CBF in SMA, IPS and FEF. To allow for a focus on cross-modal changes, the regions included in MCM and DCM analyses were Occ, FEF and IPS as obtained from the conjunction analysis (*Figure 2*). The size of these three regions was 14.4 cm$^3$ (FEF), 33.2 cm$^3$ (IPS) and 22.1 cm$^3$ (Occ), thus providing stable estimates for MCM spatial correlations between functional connectivity and glucose metabolism. The average distance between any voxel of one region to the closest voxel of another region was 46 ± 23 mm, mitigating potential cross-talk between the regions introduced through spatial smoothing.

Using BOLD-derived regions as priors for the fPET analysis may bias the direct comparison between the two imaging modalities. Hence, we also used a baseline definition for the fPET data that is independent from the BOLD data. Here, the baseline was modeled by a third-order

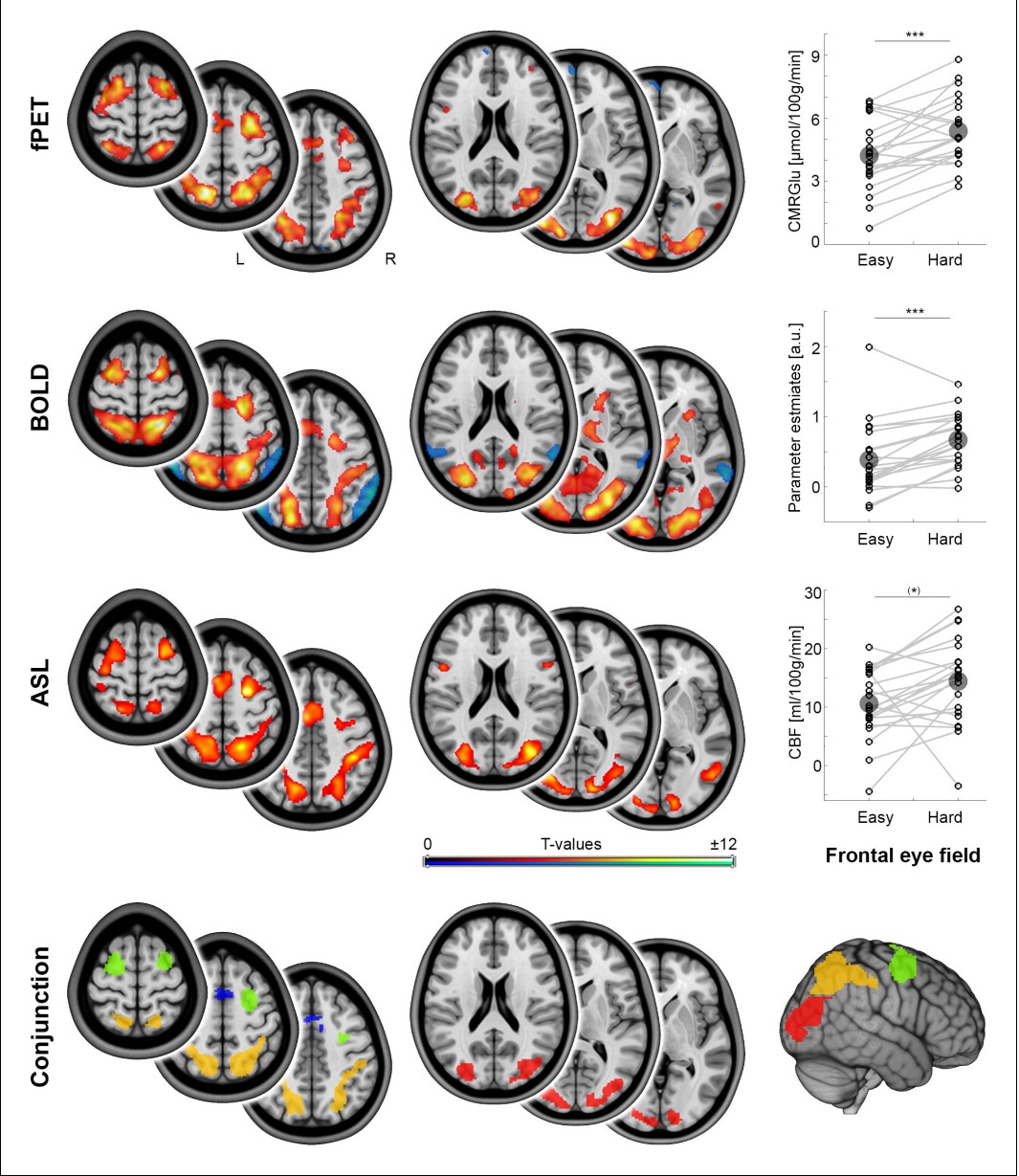

**Figure 2.** Local metabolism and neural activity. Task-specific changes of different metabolic demands as obtained with functional PET (fPET), blood oxygen level dependent (BOLD) signal and arterial spin labeling (ASL, all $p_{FWE} < 0.05$ corrected cluster level), respectively. fPET and ASL maps refer to the contrast hard > baseline, whereas BOLD maps reflect the contrast hard > control (these differ due to the manner in which the data were acquired). Dot plots depict within subject changes in glucose metabolism (CMRGlu), BOLD signal and cerebral blood flow (CBF) as a function of increasing task load in the frontal eye field (FEF, from the conjunction analysis; $^{(*)}$p = 0.07, ***p < 0.001, corrected; large circles indicate group mean values). The conjunction map shows the spatial overlap (intersection) across the three imaging modalities in the occipital cortex (red), intraparietal sulcus (orange), frontal eye field (green) and supplementary motor area (blue). Since the supplementary motor area did not show significant changes as a function of cognitive demands for CMRGlu and CBF, this region was not included in the subsequent MCM and DCM analyses. Axial slices are shown in neurological convention (left is left) at z = 0, 10 and 20 mm as well as 45, 55 and 65 mm.

The online version of this article includes the following source data and figure supplement(s) for figure 2:

**Source data 1.** Task-induced changes CMRGlu, CBF and BOLD signal.

**Figure supplement 1.** Task-induced changes in glucose metabolism obtained with an fMRI-independent baseline definition.

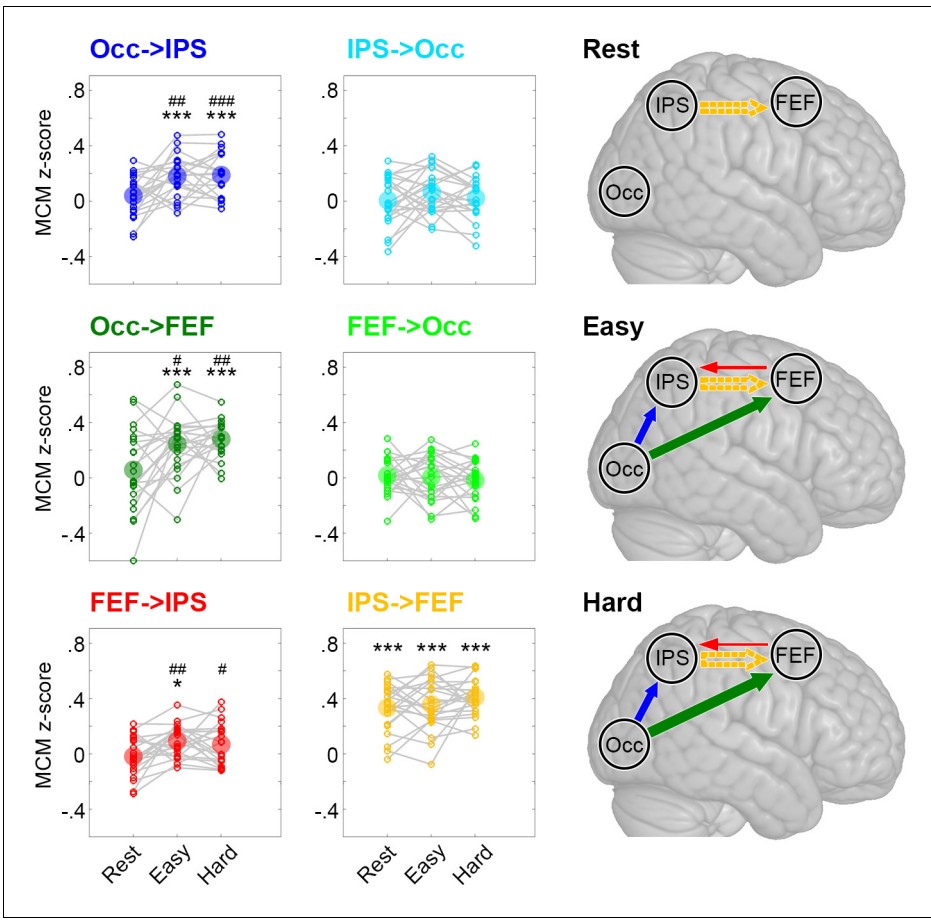

**Figure 3.** Metabolic connectivity mapping (MCM). MCM was calculated as correlation between glucose metabolism and functional connectivity across all subjects (z-transformed Pearson's r values). The small circles in the plots show z-scores of individual subjects and the big semitransparent circles indicate group mean values. At rest, only the link IPS -> FEF showed a significant correlation between metabolism and functional connectivity (orange). This correlation remained stable across all levels of task load (easy and hard). Crucially, we observed a marked increase in the correlations between metabolism and functional connectivity during task performance for Occ -> IPS (dark blue), Occ -> FEF (dark green) and FEF -> IPS (red). Changes in task load only altered the strength of these associations but not the overall pattern of interactions within the network. The resulting models for the three different conditions are schematically shown in the right panel, with the arrow thickness being proportional to the correlation strength. Solid and dashed arrows indicate connections with and without task modulation, respectively. FEF: frontal eye field, IPS: intraparietal sulcus, Occ: occipital cortex. Significant differences compared to zero *p < 0.05, **p < 0.01, ***p < 0.001 and compared to rest #p < 0.05, ##p < 0.01, ###p < 0.001, all corrected for multiple comparisons using the Bonferroni-Holm procedure.

The online version of this article includes the following source data and figure supplement(s) for figure 3:

**Source data 1.** Metabolic connectivity mapping data.
**Figure supplement 1.** Comparison between functional connectivity (FC) and metabolic connectivity mapping (MCM, as shown in *Figure 3*).
**Figure supplement 1—source data 1.** Functional connectivity data.
**Figure supplement 2.** Impact of functional connectivity preprocessing on metabolic connectivity mapping (MCM).
**Figure supplement 2—source data 1.** Metabolic connectivity mapping data with alternative functional connectivity preprocessing.
**Figure supplement 3.** Influence of spatial smoothing on MCM estimates.
**Figure supplement 3—source data 1.** Unsmoothed random permutation MCM data for IPS -> Occ.
**Figure supplement 3—source data 2.** Smoothed random permutation data for IPS -> Occ.
**Figure supplement 3—source data 3.** Unsmoothed random permutation MCM data for FEF -> Occ.
**Figure supplement 3—source data 4.** Smoothed random permutation data for FEF -> Occ.

polynomial, while modeling the task effects as nuisance variables (*Hahn et al., 2016*). Calculation of CMRGlu changes independent from the BOLD data showed similar but slightly more liberal task-specific changes in CMRGlu (*Figure 2—figure supplement 1*). The overlap between CMRGlu and BOLD-derived changes was comparable to that detected in the original analysis (Dice coefficient = 0.53).

## Task-specific association between glucose metabolism and functional connectivity

We next used MCM to assess the coupling between metabolic processes and BOLD-derived functional connectivity in the key task-related brain regions identified by our previous local analysis. At rest, significant associations between patterns of CMRGlu and functional connectivity were only observed for the link from IPS to the FEF ($p_{Bonf-Holm}$ = $10^{-7}$, *Figure 3*). This relationship was stable across the two task conditions ($p_{Bonf-Holm}$ = $10^{-11}$ to $10^{-7}$), supporting the notion that the functional interplay between IPS and FEF reflects an intrinsic and task-invariant propriety of brain organization (*Fox et al., 2006*).

Engagement in the cognitive task showed an interaction effect of condition X connection (p = 0.016) and a main effect of condition for Occ -> IPS, Occ -> FEF and FEF -> IPS (all $p_{Bonf-Holm}$ = 0.005 to 0.026). Post-hoc paired t-tests revealed that task performance induced increases in the associations between CMRGlu and functional connectivity patterns for Occ -> IPS, Occ -> FEF and FEF -> IPS connections as compared to rest (all $p_{Bonf-Holm}$ = 0.0004 to 0.02, *Figure 3*). This result highlights that task performance was specifically supported by interactions from the occipital cortex to IPS, either directly or indirectly via the FEF (*Figure 3*). A putative mediation effect of FEF was suggested by the observation that Occ -> IPS MCM values decreased in the easy condition (from 0.17 to 0.14, $p_{Bonf-Holm}$ = 0.017) when controlling for the indirect pathway (Occ -> FEF -> IPS).

Notably, functional connectivity was also sensitive to task performance with significant increases from rest to task for all three connections (*Figure 3—figure supplement 1*). Taking into account the underlying metabolic demands using MCM allowed us to infer directionality on these task-induced changes in functional connectivity, that is the influence one region exerts on another (Occ -> IPS, Occ -> FEF), as well as a separation of task effects (FEF -> IPS) from intrinsic connectivity (IPS -> FEF).

We further sought to exclude a potential influence of the preprocessing order of functional connectivity (*Carp, 2013*) and the subsequent MCM estimates. Thus, functional connectivity was also computed with an approach that includes all processing steps (regression against nuisance variables, filtering, motion scrubbing) in a single calculation using one large regression matrix (*Hallquist et al., 2013*). These analyses yielded similar MCM values as compared to the original computation (*Figure 3—figure supplement 2*).

A control analysis was performed to assess the impact of spatial smoothing on the MCM estimates (*Figure 3—figure supplement 3*). As expected, spatial correlation of randomly permuted voxels on unsmoothed data yielded MCM values close to zero (z = 0.00005 ± 0.004). Smoothing still yielded average correlations around zero but with a slightly higher variance (z = −0.001 ± 0.03). This suggests that smoothing does not systematically inflate the MCM estimates, which is also in line with our observation that half of the MCM values were approximately zero despite using smoothed data (*Figure 3*).

## Relationship between metabolic connectivity mapping and effective connectivity

We employed DCM to formally assess the association between MCM and neural dynamics. To achieve this, we constructed a space of all possible directed (effective) connections between key task-related regions (*Figure 4—figure supplement 2*). Bayesian model selection indicated that the most plausible model of neural interactions across both task conditions (easy and hard) converged with the one identified by MCM (posterior probability of 0.69 *vs.* 0.13 for the 2[nd] best model, *Figure 4*). To assess the specificity of this result, we also evaluated the relevance of each connection individually using family-wise inference. For a certain connection, all models comprising a task modulation were compared to those models without such modulation. Results showed high probabilities (posterior probability >0.99) for the majority of task-specific modulations (Occ -> FEF, FEF -> IPS)

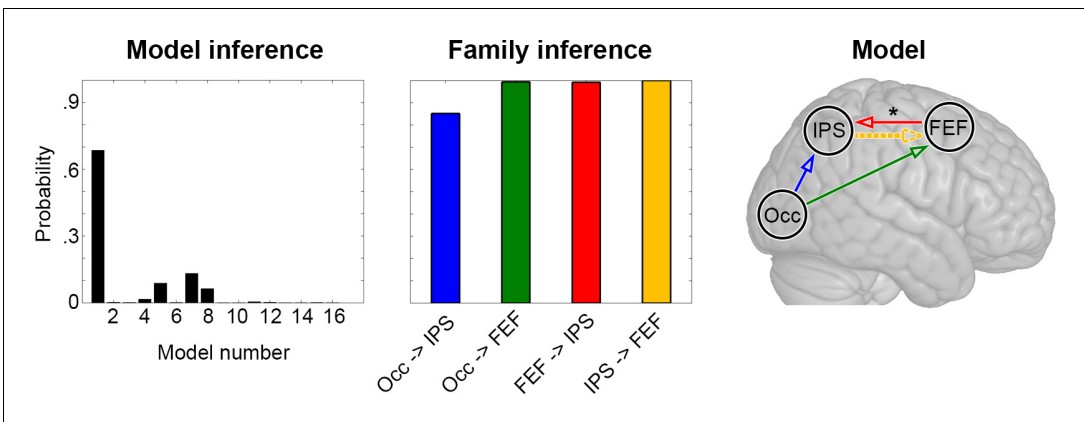

**Figure 4.** Dynamic causal modeling (DCM). Inference across the entire model space indicates the highest probability for the first model, which corresponds to the model resulting from the MCM analysis (*Figure 3*). Testing each task modulation and effective connection with family inference confirms the relevance of the individual links, with posterior probabilities close to one for all connections with the exception of Occ -> IPS. A modulation of effective connectivity as a function of task load was observed for the connection FEF -> IPS (*p < 0.05). See *Figure 3* for abbreviations and color code of connections as well as *Figure 4—figure supplement 2* for model space.

The online version of this article includes the following source data and figure supplement(s) for figure 4:

**Source data 1.** Probability of DCM models.
**Source data 2.** Probability of DCM family inference.
**Figure supplement 1.** Dynamic causal modeling (DCM) regions.
**Figure supplement 2.** Dynamic causal modeling (DCM) space.
**Figure supplement 3.** DCM results with a region of interest size of 8 mm.
**Figure supplement 3—source data 1.** Probability of DCM models with 8 mm smoothing.
**Figure supplement 3—source data 2.** Probability of DCM family inference with 8 mm smoothing.

and connections (IPS -> FEF). An exception to this was observed for Occ -> IPS linkage (posterior probability = 0.85, *Figure 4*). The reduced probability for task-based effective connectivity from occipital (Occ) to parietal (IPS) cortices mirrors the mediation effect of FEF on this connection observed with MCM (previous section). Finally, a contextual modulation of task-specific effective connections as a function of cognitive load was observed for the link FEF -> IPS (p = 0.027). Notably, the DCM results did not change when using a larger ROI size of 8 mm (*Figure 4—figure supplement 3*).

## Discussion

Using simultaneous PET/MR imaging and multimodal brain network analyses, we studied interdependent changes in neural activity, neurovascular coupling and metabolic demands as a function of cognitive task engagement and load. In addition to increments in regional neural activation and energy consumption, the execution of a cognitively challenging task elicited a marked increase in the association between glucose metabolism and functional network dynamics. In our task, requiring rapid and complex visuo-spatial reasoning, this effect was specific to feedforward connections linking the visual and dorsal attention systems. Once established, the task-based association between metabolic and neural demands reached a plateau and only few effective connections were further modulated by additional cognitive load. These findings highlight the key role of metabolic factors in supporting brain network reconfigurations as a function of cognitive engagement (*Bassett et al., 2011*; *Cocchi et al., 2014*; *Cole et al., 2013*; *Hearne et al., 2017*).

Task performance led to regional increases in different parameters of energy consumption including CMRGlu, CBF and BOLD-inferred activation. To date, only a few studies using [18F]FDG fPET imaging have assessed the possible functional relationships between these diverse parameters of brain activity (*Horwitz et al., 1995*). For example, the association between task-induced BOLD signal changes and glucose metabolism (but not CBF) has been previously studied using simple visual

stimulation and finger-tapping (*Rischka et al., 2018*) and with a visual-perceptual task (*Jamadar et al., 2019*). Here we extend upon previous work by providing a comprehensive assessment of the neural and metabolic processes supporting performance in a complex cognitive visuo-motor task. We demonstrated that the technique of fPET is sufficiently sensitive to capture metabolic differences between multiple levels of cognitive load. Moreover, our results directly support previous findings, suggesting that regional neural and metabolic changes converge in spatially circumscribed and functionally specialized brain regions (*Jamadar et al., 2019*; *Riedl et al., 2014*; *Rischka et al., 2018*).

Our results provide novel information regarding the link between energy demands and neural dynamics on a whole-brain systems level. We demonstrated that changes in metabolic demands and large-scale network dynamics converge when participants engage in a cognitive task. In other words, the association between glucose metabolism and BOLD-derived functional connectivity strengthened during cognitive performance, compared to a state of rest.

The relationship between blood flow and metabolic factors has been demonstrated to be in balance at rest, whereas task-induced neural activations induce a mismatch (*Gusnard et al., 2001*; *Raichle et al., 2001*). Specifically, CMRGlu and CBF markedly increase compared to oxygen consumption in response to increases in local neural activity. The resulting difference between CBF and oxygen consumption is thought to be the major driver of changes in the BOLD signal (*Fox and Raichle, 1986*; *Raichle, 1998*). Increases in CBF are linked to glutamate release via both neuronal and astrocytic pathways (*Attwell et al., 2010*; *Mishra et al., 2016*), supporting the contribution of glutamate-mediated post-synaptic processing to BOLD signal changes (*Logothetis et al., 2001*). The coupling between BOLD changes, local high frequency neurophysiological activity and function (*Engell et al., 2012*) also implicates the contribution of local recurrent feedback to BOLD, as well as activity reflecting efferent output (*Siero et al., 2013*). On the other hand, the relationship between CBF and CMRGlu is attributable to the downstream effects of glutamate release associated with neuronal activation. In fact, glutamate release also leads to an increased glucose uptake into neurons (*Lundgaard et al., 2015*) and astrocytes (*Zimmer et al., 2017*), to provide energy for the reversal of ion gradients and glutamate recycling (*Harris et al., 2012*; *Magistretti and Allaman, 2015*; *Raichle and Mintun, 2006*). These considerations highlight the linked but distinct mechanisms underpinning BOLD fMRI and [$^{18}$F]FDG PET signals. The asymmetry of these two different imaging techniques motivates the assumptions underlying MCM. Although the physiological mechanisms coupling CBF and CMRGlu most likely remain the same, our findings indicate that the time-dependent, nonlinear interactions between synaptic processing, energy consumption and neurovascular responses across macroscopic brain regions become apparent when engaging in a cognitively demanding task. The ability of functionally specialized brain regions to dynamically change their connectivity patterns is essential for cognitive processing (*Cocchi et al., 2013*; *Cole et al., 2013*; *Hearne et al., 2019*). Our results extend this knowledge by highlighting that external task engagement is accompanied by the convergence between metabolic factors and functional brain network reconfigurations.

In a state of rest, the association between glucose metabolism and functional connectivity was confined *within* a given functional system, namely the dorsal attention network. In contrast, task performance elicited the emergence of marked interactions *between* brain regions known to be part of distinct resting-state networks including the visual and dorsal attention systems. These observations support previous results on the emergence of between-system interactions during visual task execution (*Riedl et al., 2016*). In contrast to the substantial changes associated with the transition from rest to task, we observed that additional cognitive load does not substantially influence the association between metabolic and neural processes (*Figure 3*). These findings suggest that departing from the intrinsic (resting-state) network configuration carries the bulk of the metabolic cost associated with external task engagement. Considering that the availability of glucose in the brain is carefully maintained (unless experimentally or pathologically altered) (*Dienel, 2019*), our results put forward the hypothesis that the functional limits of efficient cognitive performance are largely related to neural processes supporting dynamic interactions between remote brain regions. Our work provides a strong motivation for future studies aiming to test this hypothesis and assess to what extent cognitive limits may depend on the efficiency of neural ensembles to use the underlying metabolic resources.

Within the task-based network (*Figures 3* and *4*), MCM and DCM allowed inferences regarding the directionality of functional connectivity between brain regions supporting task processes. Specifically, the use of these techniques draws upon metabolic knowledge (MCM) and computational modeling (DCM) to disambiguate the task-related changes in local engagement from inter-regional interactions, which compound classic linear measures of functional connectivity (*Cole et al., 2019*). Our results showed that task engagement was associated with the emergence of distinct patterns of effective connectivity from the visual to the dorsal attention networks. DCM further highlighted fine-grained modulations of specific connections as a function of increased task load. The observed bottom-up processing of visual input is in line with the role of attention networks in stimulus-driven control of attention (*Corbetta and Shulman, 2002*; *Vossel et al., 2014*) as well as the observed emergence of feedforward interaction with the FEF following stimulation of the visual cortex via transcranial magnetic stimulation (*Castrillon et al., 2020*; *Cocchi et al., 2016*). On the other hand, the lack of feedback control of the dorsal attention network over visual areas is consistent with studies showing that top-down modulation only emerges when distractors need to be filtered or context specific information extracted (*Corbetta and Shulman, 2002*; *Vossel et al., 2014*), which was not the case in our task.

Estimating measures of directed functional connectivity is a considerable challenge, particularly when those measures depend purely upon conditional likelihoods in the data (*Smith et al., 2011*), or in networks with closed cycles (*Ramsey et al., 2010*). In this context, it is worth considering the complementary nature of the two different approaches (DCM and MCM) used in this work to infer causal directionality. Whereas DCM requires careful a priori selection of regions and specification of a model space, MCM is less vulnerable to these issues (*Riedl et al., 2016*). MCM represents a simplified index of effective connectivity that draws upon the coupling between glutamate-mediated changes in the BOLD signal and metabolic demands. On the other hand, DCM provides a formal assessment of interactions between neural populations using a computational model that has been extensively validated (e.g., *Stephan et al., 2008*). By using a generative model and an observation (HRF) equation, DCM also avoids reliance upon making direct inference on conditional dependences in data, a challenge for many measures of directed functional connectivity (*Smith et al., 2011*).

In interpreting our findings, we acknowledge that the limited sample size of this study may have diminished the ability to detect more subtle changes in the coupling between metabolic and neural processes. Although small changes between the two task conditions may indeed be present or further brain regions may have been involved, these effects more likely play only a minor part in processing of the employed task and are therefore unlikely to substantially change the implications of our results. Also, our task only tested for a circumscribed set of visual-motor cognitive domains. While it is unlikely that the observed effects are specific to a given set of cognitive functions, future work is required to expand current findings to different domains. By successfully linking metabolic and functional network dynamics, the current study provides new methodological and neurobiological frameworks to understand the nature of pathological alterations of brain functions. Specifically, our paradigm and findings may help unfold the complex interdependent effects of cognitive, metabolic, and neural factors in pathologies such as epilepsy, schizophrenia and Alzheimer's disease (*Scherr et al., 2019*).

## Materials and methods

### Experimental design

All participants underwent a single PET/MRI examination during performance of a challenging cognitive task within this cross-sectional single-site study. The experimental sequence comprised a structural scan (8 min), ASL and BOLD signal acquisitions as well as simultaneous fPET imaging at rest and during task performance (*Figure 1A*). During periods of rest, participants were instructed to look at a crosshair, relax and not focus on anything in particular. First, one ASL and BOLD sequence (6 min each) were each obtained at rest. Next, fPET started with a baseline of 8 min, which was followed by 4 periods of continuous task performance (6 min each, two easy, two hard, randomized) with rest periods following all tasks (5 min each). During the task periods (see description below), ASL and BOLD data were also acquired in pseudorandom order. BOLD data obtained during these continuous task periods were used for the computation of functional connectivity. The session

finished with the acquisition of BOLD data in a conventional block design with performance of the same task (12 task blocks, 30 s each, four easy, four hard, four control, randomized, 8.17 min in total) separated by short periods of rest (10 s each). Data from this BOLD block design allowed inference of task-specific changes in neuronal activation and dynamics using DCM. Thus, participants completed the same task in both the continuous task performance and block design components of the session. The timing of these tasks differed in order to optimize the efficiency of the various analyses (cross-modal functional connectivity *vs.* activation analyses, respectively; see *Figure 1C*). The total scan time was 100 min, which represents a typical duration for PET studies. The use of simultaneous PET/MR enabled the acquisition of all data at rest and during task performance in a single scan session, thus decreasing intrasubject variability, mitigating performance differences and practice effects across fPET and fMRI acquisitions.

## Cognitive task

An adapted version of the video game Tetris was implemented with two levels of cognitive load (easy and hard, *Figure 1B*). Bricks descended from the top of the screen and required rotation and alignment in order to build horizontal lines. Completed lines then vanished and increased the score. The upcoming brick was shown in a preview on the screen. To move the bricks participants used four buttons with the right hand only (index finger: move brick left, middle: rotate, ring: down, small: right). The two task conditions were different in the speed of the descending bricks (easy/hard: 1/3 lines per sec) and the number of incomplete lines that were already built at the bottom (easy/hard: 2/6 lines out of 20). An additional control condition was used for the BOLD block design only (*Figure 1B*). Here, bricks required navigation through a channel, which then vanished at the bottom (i.e., there was no possibility to build or complete lines). To familiarize with the task and the control buttons, participants completed a 30 s training of each condition before the scan started. The implementation represents a cognitively challenging task which requires rapid visuo-spatial motor coordination, spatial planning and problem solving. The adopted task is also well-suited for continuous performance of several minutes, which is required for the detection of task-specific glucose metabolism with fPET (*Rischka et al., 2018*).

## Participants

Twenty-two healthy participants were recruited for this study (mean age ± SD = 24.0 ± 3.1 years, 11 female, all right-handed). The sample size is based on that of the original study introducing the MCM approach which showed robust associations between glucose metabolism and functional connectivity (*Riedl et al., 2016*). All participants underwent a standard medical examination at the initial screening visit, which included blood tests, electrocardiography, neurological testing and the Structural Clinical Interview for DSM-IV performed by an experienced psychiatrist. Female participants also underwent a urine pregnancy test at the screening visit and before the PET/MRI scan. Exclusion criteria were current and previous (12 months) somatic, neurological or psychiatric disorders, current and previous substance abuse or psychotropic medication, contraindications for MRI scanning, pregnancy or breast feeding, previous study-related radiation exposure (10 years) and previous experience with Tetris in the last 3 years. After detailed explanation of the study protocol, all participants gave written informed consent. The study was approved by the Ethics Committee (ethics number: 1479/2015) of the Medical University of Vienna and procedures were carried out in accordance with the Declaration of Helsinki.

## PET/MRI data acquisition

Synthesis of the radiotracer [$^{18}$F]FDG and PET acquisition was carried out as described previously (*Rischka et al., 2018*). The radiotracer was injected via the cubital vein as bolus (510 kBq/kg/frame, 1 min) plus constant infusion (40 kBq/kg/frame, 51 min) using a perfusion pump (Syramed μSP6000, Arcomed, Regensdorf, Switzerland), which was kept in an MRI-shield (UniQUE, Arcomed).

The structural MRI was acquired with a T1-weighted MPRAGE sequence (TE/TR = 4.21/2200 ms, TI = 900 ms, flip angle = 9°, matrix size = 240×256, 160 slices, voxel size = 1×1 x 1 mm + 0.1 mm gap, 7.72 min). ASL was obtained with a 2D pseudo-continuous ASL sequence (TE/TR = 12/4060 ms, post label delay = 1800 ms, flip angle = 90°, matrix size = 64×64, 20 slices, voxel size = 3.44×3.44 x 5 mm + 1 mm gap, 6 min) (*Kilroy et al., 2014*). All BOLD data were acquired using an EPI sequence

(TE/TR = 30/2000 ms, flip angle = 90°, matrix size = 80×80, 34 slices, voxel size = 2.5×2.5 x 2.5 mm + 0.825 mm gap).

## Blood sampling

Before the PET/MRI measurement the individual blood glucose level was determined as triplicate ($Glu_{plasma}$). During the scan manual arterial blood samples were drawn at 3, 4, 5, 14, 25, 36 and 47 min after start of the tracer application. This sampling scheme has been shown to be sufficient for the determination of the input function for the employed bolus plus infusion protocol (*Rischka et al., 2018*). Processing of blood samples included measurement of whole-blood activity in a gamma-counter (Wizward[2], 3', Perkin Elmer), separation of plasma and again measurement of plasma activity. For the arterial input function, manual samples were linearly interpolated to match PET frames and multiplied by the average plasma-to-whole-blood ratio.

## Quantification of glucose metabolism (CMRGlu)

PET image reconstruction, preprocessing and quantification were carried out as described previously (*Rischka et al., 2018*). In short, PET data were reconstructed to frames of 30 s (matrix size = 344×344, 127 slices) and corrected for attenuation using an established database approach (*Burgos et al., 2014*). Image preprocessing in SPM12 included motion correction (quality = 1, registered to mean), spatial normalization via the T1-weighted structural MRI and smoothing with an 8 mm Gaussian kernel. Data were masked to include only gray matter voxels and a low-pass filter was applied with the cutoff frequency set to half the task duration (i.e., 3 min). To separate task effects from baseline metabolism, a general linear model was used with four regressors representing the baseline, the task conditions (easy and hard separately, linear ramp function with slope = 1 kBq/frame) and head movement (first principal component of the six motion regressors). The baseline was defined as average time course of gray matter voxels which do not change in the hard condition (*vs.* baseline) of the corresponding individual BOLD block design ($p_{FWE} < 0.05$ corrected voxel level, see below). The approach was chosen to capture the maximum amount of voxels that are changed by the task and since this was previously shown to provide good model fits and a robust estimate of baseline CMRGlu (*Rischka et al., 2018*). Finally, the Patlak plot was applied to compute the influx constant $K_i$ for baseline and task effects separately, which was then converted to the cerebral metabolic rate of glucose (CMRGlu) as

$$CMRGlu = K_i^* Glu_{plasma}/LC^* 100, \qquad (1)$$

with LC being the lumped constant = 0.89. Thus, the task effects represent CMRGlu that is consumed on top of the baseline CMRGlu. At the group level, the effects of cognitive load (easy and hard) were separately assessed using a one-sample t-test ($p_{FWE} < 0.05$ corrected cluster level, height threshold of $p < 0.001$ uncorrected voxel level).

## Quantification of cerebral blood flow (CBF)

Processing of pCASL data was carried out according to standard procedures (*Wang et al., 2005*). First, to reduce spurious scanner effects, voxels with signal intensity < 0.8 * mean value were set to 0 (similar to fMRI analysis in SPM). After motion correction in SPM12, the equilibrium magnetization of the brain $M_0$ was computed as the average of all non-labeled images. Non-brain voxels were then removed by applying the brain extraction tool (*Smith, 2002*) to the $M_0$ image and masking all other images. CBF was quantified with the following equation:

$$CBF = \frac{MR_{1a}}{2M_0\{exp(R_{1a})\, exp[(+)R_{1a}]\}} \qquad (2)$$

with ΔM being the difference between label and control images, λ = 0.9 g/ml the blood/tissue water partition coefficient, $R_{1a}$ = 0.6 s$^{-1}$ the longitudinal relaxation rate of blood, α = 0.8 the tagging efficiency, ω the post-labeling delay corrected for slice timing differences and τ = 1508 ms the duration of the labeling pulse. CBF was then averaged across time separately for each condition (rest, easy, hard). The resulting CBF maps were then spatially normalized via the T1-weighted structural image and smoothed with an 8 mm Gaussian kernel. Because the resulting maps linked to task conditions represent the sum of both baseline and task effects, the baseline CBF was subtracted to obtain the

pure easy and hard CBF maps. Task-specific effects were evaluated at the group level using a one-sample t-test ($p_{FWE}$ < 0.05 corrected cluster level, height threshold of p < 0.001 uncorrected voxel level).

## BOLD changes

Task-induced changes in BOLD signal were estimated using data from the block design acquired in the final part of the experiment (*Figure 1A*, red box). Data preprocessing was carried out using SPM12 as described previously (*Rischka et al., 2018*). After correction of slice timing differences (reference: middle slice) and motion (quality = 1, registered to mean) the data were normalized to MNI-space and smoothed with an 8 mm Gaussian kernel to match the kernel used for the PET and ASL data. To estimate task-specific effects the general linear model was used with one regressor for each condition (easy, hard, control) and nuisance regressors for movement, white matter and cerebrospinal fluid. The first level contrasts of interest were easy *vs.* control and hard *vs.* control, which were then carried over to a second level random effects analysis. At the group level, first level contrasts were entered in a one-sample t-test ($p_{FWE}$ < 0.05 corrected cluster level, height threshold of p < 0.001 uncorrected voxel level).

## Conjunction of energy demands

To provide a robust identification of the network involved in task performance, the different metabolic demands reflected by regional CMRGlu, CBF and BOLD signal changes were combined. For each of these imaging modalities the group-average task effects were obtained separately by a one-sample t-test ($p_{FWE}$ < 0.05 corrected cluster level, height threshold of p < 0.001 uncorrected voxel level). The resulting patterns of suprathreshold effects were binarized and a conjunction map across the three imaging modalities was calculated based on the intersection among the binary clusters (*Figure 2*). Brain regions included in this conjunction map were used for the subsequent MCM analysis with a focus on large, symmetric regions (except the SMA, see results). Homologous regions in both hemispheres were combined since the dorsal attention network is not considered to be strongly lateralized (*Fox et al., 2006*).

## Metabolic connectivity mapping (MCM)

Proceeding from the network defined above, we assessed the relationship between functional connectivity and glucose metabolism in the three conditions (rest, easy, hard) using metabolic connectivity mapping (MCM *Riedl et al., 2016*). MCM evaluates the correspondence between *regional* patterns of BOLD-derived functional connectivity and glucose metabolism. As the majority of energy is consumed post-synaptically, the method further allows inference on the direction of functional connectivity between any two brain regions.

MCM involves a seed-to-voxel *temporal* correlation between region A (mean BOLD signal of the seed) and region B (BOLD signal of each voxel). This yields a pattern of functional connectivity values in B reflecting the voxel-wise connectivity with the mean signal in A. The connection is assumed to have directionality from region A to B when the pattern of voxel-wise functional connectivity in B shows a significant *spatial* correlation with the corresponding voxel-wise pattern of glucose metabolism in region B. The main assumption of the approach is that the majority of energy demands arise post-synaptically, that is, in the target region (*Attwell and Laughlin, 2001*; *Harris et al., 2012*; *Mergenthaler et al., 2013*; *Riedl et al., 2016*), which are tightly linked to post-synaptic glutamate-mediated processes (*Attwell et al., 2010*; *Mishra et al., 2016*). Although BOLD signal changes receive a substantial drive from post-synaptic activity from distant inputs (*Logothetis et al., 2001*), it also receives contributions from local recurrent feedback and output (spiking)-related activity. Hence, the BOLD signal strongly covaries with local high frequency neurophysiological activity (*Engell et al., 2012*), as well as inputs to sensory cortex (*Aquino et al., 2012*) and behavioral output from primary motor cortex (*Siero et al., 2013*). MCM exploits this asymmetry across data modalities: If the BOLD-derived pattern of functional connectivity in region B is indeed caused by the influence region A exerts on region B (i.e., testing directionality of pairwise interactions), this will yield a corresponding pattern of the underlying energy consumption in region B, given the tight coupling between BOLD signal changes and glucose metabolism.

Functional connectivity was computed at rest (i.e., initial 6 min period, *Figure 1A*, orange box) and for continuous task performance (6 min task periods) (*Hahn et al., 2018*). Data preprocessing was identical to that of the BOLD block design data up to (including) spatial smoothing. To mitigate against the confounding effect of task-related head movement, motion scrubbing was applied (*Power et al., 2015*). Here, the framewise displacement was calculated as the sum of frame-to-frame absolute differences across all six motion parameters. Rotation parameters were converted from degrees to millimeters by calculating the displacement on the surface of a sphere (radius = 50 mm). All frames with framewise displacement > 0.5 mm (plus one frame back and two forward) were removed from the further analysis (5 ± 8% of frames across all conditions). Afterwards, linear regression against confounding signals (six movement parameters, white matter, cerebrospinal fluid) and bandpass filtering was applied. Cut-off frequencies were chosen to enable comparison of functional connectivity at resting-state and during task performance (0.01 Hz < f < 0.15 Hz) (*Sun et al., 2004*). For MCM, functional connectivity was computed as seed-to-voxel correlations between two brain regions followed by z-transformation. In other words, the average time course of region A was correlated with each voxel's time course of region B, yielding a voxel-wise pattern of functional connectivity in region B. For direct comparison between MCM and functional connectivity, we also report the functional connectivity as the Pearson's correlation between time courses of the three brain regions.

MCM was then computed as linear spatial correlation (z-transformed Pearson's r) between regional patterns of functional connectivity and glucose metabolism of region B (i.e., between the vector of voxel-wise functional connectivity and the vector of voxel-wise CMRGlu values). Note that MCM is a directed measure, in the sense that two regions with a symmetric functional connectivity will yield MCM estimates with directed links towards those regions, where spatial patterns of functional connectivity and metabolism show a significant association. The procedure was repeated for all six combinations of regions of the network identified above, which results in an asymmetric (directional) $3 \times 3$ matrix of connections.

We tested for a potential mediation effect of FEF on the connection Occ -> IPS by adapting an established statistical approach (*Baron and Kenny, 1986*) to the MCM framework. Specifically, a partial correlation between BOLD functional connectivity and glucose metabolism was calculated in the IPS, while correcting for the BOLD functional connectivity between IPS and FEF. The resulting MCM values were then compared to the original MCM values of Occ -> IPS using a paired t-test.

To assess the influence of spatial smoothing on MCM estimates, a control analysis was carried out testing whether a spatial correlation of zero is artificially inflated by smoothing. For each subject, unsmoothed BOLD functional connectivity and CMRGlu data were separately permuted across voxels. MCM was then calculated before and after spatial smoothing (8 mm) using the identical regions of interest as for the main analysis. To generate a null distribution representing the influence of smoothing in these finite sampled data, this procedure was repeated 500 times for each of the 22 subjects.

## Dynamic causal modeling (DCM)

The directed functional connectivity model obtained from the MCM analysis was benchmarked against directed effective connectivity inferred from DCM. DCM for fMRI couples a simplified (bilinear) model of neural dynamics with a biophysical model of hemodynamics to infer causal interactions between remote brain regions (*Friston et al., 2003*; *Stephan et al., 2010*). Regional BOLD time courses were extracted from each brain region of the above defined network using fMRI data derived from the task blocks (*Figure 1A*, red box). Regions of interest (ROIs) were defined from peak signals at group level ($p_{FWE}$ < 0.05 corrected voxel level). For each ROI, the group-level coordinate was individually shifted over subject's nearest maximum effect size (F-test across all three conditions, *Figure 4—figure supplement 1*) within the conjunction map defined by our previous analysis step (*Figure 2*). The BOLD signal was extracted as the eigenvariate within a sphere of 5 mm radius, adjusted for the effect of interest. Control analyses were performed using a sphere of 8 mm radius. Thus, the region employed for DCM represents a subregion of that used for MCM, with a focus on the individual location of task-specific changes in the BOLD signal. DCM is a hypothesis-driven framework that requires the definition of a model space given by: (i) intrinsic (context-independent) connections, (ii) task-driven modulation of these intrinsic connections and (iii) extrinsic inputs (here, the visual stimuli). All tested models had extrinsic inputs to the occipital cortex but differed regarding the absence or presence of intrinsic connections and task modulations (*Figure 4—figure*

*supplement 2*). For each of these models a bilinear DCM was estimated. Model comparison was performed using random effects Bayesian model selection. Model selection rests upon identifying the model that has the highest likelihood, conditioned on the observed data and penalized by relative model complexity. In the context of uniform model priors, as we implemented, model selection identifies the model with the highest posterior probability (evidence) (*Penny et al., 2007*). Each effective connection was also individually tested for the presence of task modulation using model family inference (i.e., all models comprising task modulation of a given connection against those without).

## Statistical analysis

All *p*-values were corrected for multiple comparisons using the Bonferroni-Holm procedure (number of performed tests for each separate analysis indicated below in brackets). To assess the spatial agreement of task effects between imaging modalities the Dice coefficient was used. Differences between the two task levels (easy *vs.* hard) were evaluated using two-tailed paired t-tests. This statistical analysis was applied to both behavioral data and imaging parameters of each brain region defined in the conjunction map (three parameters * four brain regions, *Figure 2*). For the MCM analysis, the significance of each connection was tested separately using one-sample t-tests against zero (six connections * three conditions). As MCM yields one spatial correlation value for each subject, the test for each connection and condition is pooled across 22 correlations, corresponding to $n = 22$ subjects. Differences of the two task levels against rest were computed step-wise. First, a repeated measures ANOVA was calculated to assess a potential interaction effect between the factors condition and connection (one interaction). Next, the main effect of condition was assessed for each connection (six connections), followed by post-hoc paired t-tests of each task level against baseline (two conditions).

## Acknowledgements

The authors are particularly grateful to Prof. Danny JJ Wang, Stevens Neuroimaging and Informatics Institute, University of South California, and the Regents of the University of California, for providing the pCASL sequence. We are further grateful to J Raitanen, J Völkle and A Pomberger for radioligand synthesis. We would like to thank K Papageorgiou and the diploma students of the Neuroimaging Labs (NIL) for medical support, S Klug for measurement support, V Ritter, K Einenkel and E Sittenberger for subject recruitment, A Jelicic for implementation of the task and MB Reed for proofreading.

This research was supported by a grant from the Austrian Science Fund to A Hahn (FWF KLI 610). L Rischka is recipient of a DOC Fellowship of the Austrian Academy of Sciences at the Department of Psychiatry and Psychotherapy, Medical University of Vienna. The scientific project was performed with the support of the Medical Imaging Cluster of the Medical University of Vienna. L Cocchi is supported by the Australian National Health Medical Research Council (LC 1099082 and 1138711).

## Additional information

### Competing interests

Wolfgang Wadsak: WW declares to having received speaker honoraria from the GE Healthcare and research grants from Ipsen Pharma, Eckert-Ziegler AG, Scintomics, and ITG; and working as a part time employee of CBmed Ltd. (Center for Biomarker Research in Medicine, Graz, Austria). Marcus Hacker: MH received consulting fees and/or honoraria from Bayer Healthcare BMS, Eli Lilly, EZAG, GE Healthcare, Ipsen, ITM, Janssen, Roche, and Siemens Healthineers. Siegfried Kasper: SK received grants/research support, consulting fees and/or honoraria within the last three years from Angelini, AOP Orphan Pharmaceuticals AG, Celgene GmbH, Eli Lilly, Janssen-Cilag Pharma GmbH, KRKA-Pharma, Lundbeck A/S, Mundipharma, Neuraxpharm, Pfizer, Sage, Sanofi, Schwabe, Servier, Shire, Sumitomo Dainippon Pharma Co. Ltd., Sun Pharmaceutical Industries Ltd. and Takeda. Rupert Lanzenberger: RL received conference speaker honorarium within the last three years from Shire and support from Siemens Healthcare regarding clinical research using PET/MR. He is shareholder of BM Health GmbH since 2019. The other authors declare that no competing interests exist.

## Funding

| Funder | Grant reference number | Author |
|---|---|---|
| Austrian Science Fund | FWF KLI 610 | Andreas Hahn |
| Austrian Academy of Sciences | DOC Fellowship | Lucas Rischka |
| National Health and Medical Research Council | 1099082 | Luca Cocchi |
| National Health and Medical Research Council | 1138711 | Luca Cocchi |

The funders had no role in study design, data collection and interpretation, or the decision to submit the work for publication.

## Author contributions

Andreas Hahn, Conceptualization, Data curation, Formal analysis, Supervision, Funding acquisition, Investigation, Visualization, Methodology, Project administration; Michael Breakspear, Conceptualization, Supervision, Investigation, Methodology; Lucas Rischka, Data curation, Formal analysis, Methodology; Wolfgang Wadsak, Conceptualization, Data curation, Supervision; Godber M Godbersen, Verena Pichler, Paul Michenthaler, Thomas Vanicek, Data curation; Marcus Hacker, Siegfried Kasper, Rupert Lanzenberger, Conceptualization, Supervision; Luca Cocchi, Conceptualization, Formal analysis, Supervision, Investigation, Visualization, Methodology

## Author ORCIDs

Andreas Hahn (iD) https://orcid.org/0000-0001-9727-7580
Michael Breakspear (iD) http://orcid.org/0000-0003-4943-3969
Luca Cocchi (iD) https://orcid.org/0000-0003-3651-2676

## Ethics

Clinical trial registration ClinicalTrials.gov ID: NCT03485066.
Human subjects: After detailed explanation of the study protocol, all participants gave written informed consent. The study was approved by the Ethics Committee (ethics number: 1479/2015) of the Medical University of Vienna and procedures were carried out in accordance with the Declaration of Helsinki.

## Decision letter and Author response

Decision letter https://doi.org/10.7554/eLife.52443.sa1
Author response https://doi.org/10.7554/eLife.52443.sa2

# Additional files

## Supplementary files

• Transparent reporting form

## Data availability

We cannot publicly share raw data for reasons of data protection; processed data supporting the results of this manuscript have been deposited to Dryad Digital Repository under the DOI https://doi.org/10.5061/dryad.zcrjdfn7p.

The following dataset was generated:

| Author(s) | Year | Dataset title | Dataset URL | Database and Identifier |
|---|---|---|---|---|
| Hahn A, Breakspear M, Rischka L, Wadsak W, Godbersen GM, Pichler | 2020 | Data from: Reconfiguration of functional brain networks and metabolic cost converge during task performance | https://doi.org/10.5061/dryad.zcrjdfn7p | Dryad Digital Repository, 10.5061/dryad.zcrjdfn7p |

V, Michenthaler P,
Vanicek T, Hacker
M, Kasper S, Lan-
zenberger R, Coc-
chi L

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
