## [Decision Letter]

**Acceptance summary:**

Hahn and colleagues analyzed data from 22 participants performing a complex cognitive task (Tetris) during concurrent fMRI-PET. The study demonstrated that performing the complex cognitive task yielded significant reconfiguration of brain connectivity (relative to rest). Yet increases in task difficulty yielded relatively small changes (compared with between rest and task). The convergence between two very different approaches – Metabolic Connectivity Mapping (based on PET-MR) and dynamic causal modeling (based on fMRI) – was impressive.

**Decision letter after peer review:**

Thank you for submitting your article "Reconfiguration of functional brain networks and metabolic cost converge during task performance" for consideration by *eLife*. Your article has been reviewed by three peer reviewers, one of whom is a member of our Board of Reviewing Editors, and the evaluation has been overseen by Timothy Behrens as the Senior Editor.

The reviewers have discussed the reviews with one another and the Reviewing Editor has drafted this decision to help you prepare a revised submission.

Summary:

In the current manuscript, Hahn and colleagues analyzed data from 22 participants in a concurrent fMRI-PET dataset. Participants were either at rest or asked to play a Tetris game at two different levels of difficulty. While playing, fMRI BOLD or ASL was collected concurrently with FDG PET imaging. In addition, a set of baseline fMRI (BOLD and ASL) resting state scans and a final set of blocked BOLD task scans were collected in the absence of PET imaging. With this data, the authors examined the overlap of glucose metabolism (PET), BOLD, and cerebral blood flow (ASL) during the task. The authors then used a combination of the PET and BOLD data to make estimates of directed connectivity during the Tetris task, a technique called Metabolic Connectivity Mapping (MCM), and compared the results of this MCM analysis to a DCM analysis on the BOLD data.

This is a unique and interesting dataset, providing the authors with concurrent measurements of both fMRI and PET measures. The first analysis (Figure 2) demonstrating substantial overlap between glucose metabolism, BOLD, and ASL appears quite compelling. The convergence between two very different approaches (DCM and MCM; Figure 3) is quite impressive.

Essential revisions:

1) Throughout the manuscript, the authors refer to BOLD measurements as "neural activation" measurements. This is imprecise terminology to use, especially in the current context, where the BOLD measurements are being contrasted with measurements of cerebral blood flow and glucose metabolism. Likely all of these measures are at some level related to neural activity, but all are indirect, and it's not clear why BOLD measurements receive a privileged status on this front in this manuscript. I would strongly recommend shifting this terminology.

2) This terminological distinction is important because it obscured the approach used in the MCM analyses in this paper. The authors argue that glucose metabolism tends to occur largely post-synaptically, and thus a comparison between BOLD functional connectivity and PET glucose metabolism profiles allows for the determination of directed information flow. But there is also substantial evidence that the BOLD signal is better related to neural post-synaptic potentials than action potentials (e.g., Logothetis et al., 2001). Thus this logic is unclear. It would be helpful if the authors could expand further on the motivation and validation for this analysis.

3) Given general issues in the field with interpreting directed connectivity (e.g., Ramsey et al., 2010; Smith et al., 2011), validation with DCM seems like an insufficient standard. The authors should be explicit about the pitfalls of both approaches (DCM and MCM) and whether they cover each other's pitfalls. For example, DCM is probably sensitive to missing nodes, while MCM won't have that issue (as far as we can tell), so the two approaches nicely complement each other in this respect. Are there pitfalls that apply to both DCM and MCM? If so, how serious are these pitfalls and can they be addressed with control analyses?

4) Given the large amount of spatial smoothing that was performed across all modalities (8 mm), how informative is a spatial correlation calculated within ROIs for MCM? One way to test for this more robustly is to perform permutation testing by comparing a BOLD connectivity map from one subject to a glucose metabolism map of a different subject (and build a null distribution through many such permutations), rather than performing a basic t-test against zero (and similarly for the comparison against rest). This will determine whether spatial similarity is truly indicative of connectivity-metabolic coupling, or whether it is observed as a function of the large smoothing kernel (and relatively small ROIs).

5) Why were DCM ROIs different from the MCM ROIs (Figure 4—figure supplement 1)? If DCM is used for validation, it seems to make sense to use the same regions in both analyses. Furthermore, how does the interaction between the size of the ROIs (5 mm) and the smoothing kernel (8mm) affect the results?

6) Some of the statistical tests are unclear. The authors wrote that "the significance of each connection was tested separately using one-sample t-tests against zero". So let's consider one connection: FEF to IPS. The MCM analysis yields one spatial correlation for each subject. So can I confirm the t-test involves 22 numbers (corresponding to the 22 subjects)? Because a p value of 10e-12 is quite impressive with only 22 subjects; one would need a t-score of around 14?

7) For Figure 3, the authors say that the thickness of arrows depend on r values. More details will be useful. For example, when looking at Occ -> FEF, the two box plots for easy and hard conditions look highly overlapping, but the green arrow was much thicker in the hard condition than easy condition. And from what I gather from the plot, the IPS->FEF during hard condition has higher MCM correlation than Occ->FEF during hard condition. So why is the green arrow thickness than the yellow arrow during the hard condition?

8) It would be important to report the pure functional connectivity among the 3 regions. More specifically, are there region pairs with strong functional connectivity, but the MCM connectivity is weak in both directions. If so, how can this be interpreted? Similarly, are there regions with weak functional connectivity but the MCM connectivity is strong? If so, how can this be interpreted?

9) The DCM modeling needs to be further elaborated. Maybe the current details are sufficient for a practiced DCM user, but as someone who is somewhat familiar with DCM (but do not actively use it), I found it hard to know how task modulation and intrinsic connection were modeled in the generative model.

10) "A putative mediation effect of FEF was suggested by the observation that Occ -> IPS MCM values decreased in the easy condition (from 0.17 to 0.14, pBonf-Holm = 0.017) when controlling for the indirect pathway (Occ -> FEF -> IPS)." – Please elaborate on how this is done.

11) Literature review and discussion.

A) Given the nature of the paper, the authors should include a discussion of how their findings compare to a large and long-standing literature on cerebral blood flow and metabolism, which explicitly contrasts cerebral blood flow, oxygen consumption, and glucose metabolism to one another in and outside of tasks contexts, e.g., from Marc Raichle and others (see for example: Gusnard and Raichle, 2001). The current work would have substantially greater impact if it were to engage more deeply with this older literature. For example, do the authors mean to suggest (as might appear from their statement in the Abstract that "dynamic regulation of neural-metabolic coupling is essential to support human cognition") that different mechanisms exist to couple neural activity with glucose consumption depending on the task at hand? This claim seems quite large and far reaching given the evidence, and if true, would have profound implications in invalidating the current use and interpretation of many neuroimaging methods.

B) The authors should also be more precise about differences with previous work (e.g., Jamadar et al., 2019; Rischka et al., 2018). For example, "Here we extend upon previous work by providing a comprehensive assessment of the neural and metabolic processes supporting performance in a complex cognitive visuo-motor task" can be briefly expanded to mention that Rischka investigated a finger-tapping task and Jamadar investigated a visual checkerboard task.

12) The Discussion states: "Our results extend this knowledge by highlighting that the convergence between metabolic factors and functional network reconfiguration is pivotal to support cognition". However, the relationship with cognition or performance was not tested here (e.g. MCM was not shown to be stronger for those subjects who performed better). As such, I would recommend adjusting this interpretation.

---

## [Author Response]

Essential revisions:1) Throughout the manuscript, the authors refer to BOLD measurements as "neural activation" measurements. This is imprecise terminology to use, especially in the current context, where the BOLD measurements are being contrasted with measurements of cerebral blood flow and glucose metabolism. Likely all of these measures are at some level related to neural activity, but all are indirect, and it's not clear why BOLD measurements receive a privileged status on this front in this manuscript. I would strongly recommend shifting this terminology.

We agree with the editors/reviewers that all three imaging modalities reflect different aspects of neural activation. Thus, when referring to BOLD measurements the term “neural activation” was changed to best match the specific context (see examples below).

Results:

*“*We first integrated information regarding task-related changes in local BOLD signals, cerebral blood flow (CBF) and glucose consumption to identify functionally segregated brain regions involved in task performance (Figure 1C). This analysis leveraged the recently introduced approach of functional PET (fPET) imaging to determine task-specific changes in the cerebral metabolic rate of glucose (CMRGlu) (Hahn et al., 2016; Rischka et al., 2018) as well as canonical analyses for BOLD and ASL signal changes.”

Results:

“Relative to resting-state, task performance elicited significant increases in CMRGlu, CBF and BOLD-signals […]”

Materials and methods:

“Data from this BOLD block design allowed inference of task-specific changes in neural activation and dynamics using DCM.”

Materials and methods:

“BOLD changes

Task-induced changes in BOLD signal were estimated using data from the block design acquired in the final part of the experiment (Figure 1A, red box). […] To estimate task-specific effects the general linear model was used with one regressor for each condition (easy, hard, control) and nuisance regressors for movement, white matter and cerebrospinal fluid.”

Materials and methods:

“To provide a robust identification of the network involved in task performance, the different metabolic demands reflected by regional CMRGlu, CBF and BOLD signal changes were combined.”

Figure 1:

“In the final part of the experiment, participants performed 12 task blocks (easy, hard and control, 30 s for each block, red) during BOLD acquisition to estimate neural task effects and to compute effective connectivity using dynamic causal modelling. […]

First, brain regions involved in task processing were identified as conjunction of task-specific changes in glucose metabolism (fPET), blood oxygenation (BOLD) and blood flow (ASL).”

Figure 2:

“Dot plots depict within subjects changes in glucose metabolism (CMRGlu), BOLD signal and cerebral blood flow (CBF) as a function of increasing task load in the frontal eye field (FEF, from the conjunction analysis; ^(*)^p = 0.07, ***p < 0.001, corrected; large circles indicate group mean values).”

2) This terminological distinction is important because it obscured the approach used in the MCM analyses in this paper. The authors argue that glucose metabolism tends to occur largely post-synaptically, and thus a comparison between BOLD functional connectivity and PET glucose metabolism profiles allows for the determination of directed information flow. But there is also substantial evidence that the BOLD signal is better related to neural post-synaptic potentials than action potentials (e.g., Logothetis et al., 2001). Thus this logic is unclear. It would be helpful if the authors could expand further on the motivation and validation for this analysis.

While it is true that post-synaptic responses make a strong contribution to both glucose metabolism and BOLD responses, considerable work since the seminal paper by Logothetis et al. has shown that BOLD reflects more of a composite of local activity which includes post-synaptic activity arising from distant inputs, but also local recurrent feedback and output-related activity. The latter can be seen in studies that show a link between BOLD responses in M1 and motor movement rates. We have expanded on the explication of MCM and also disambiguated between fPET and BOLD signal changes):

“The main assumption of the approach is that the majority of energy demands arise post-synaptically, i.e., in the target region (Attwell and Laughlin, 2001; Harris et al., 2012; Mergenthaler et al., 2013; Riedl et al., 2016) which are tightly linked to post-synaptic glutamate-mediated processes (Attwell et al., 2010; Mishra et al., 2016). […] MCM exploits this asymmetry across data modalities: If the BOLD-derived pattern of functional connectivity in region B is indeed caused by the influence region A exerts on region B (i.e., testing directionality of pairwise interactions), this will yield a corresponding pattern of the underlying energy consumption in region B, given the tight coupling between BOLD signal changes and glucose metabolism.”

We have also significantly extended the Discussion on the underlying physiological mechanisms of BOLD and [^18^F]FDG PET imaging (response to main issue 11A) and added the following text:

“The relationship between blood flow and metabolic factors has been demonstrated to be in balance at rest, whereas task-induced neural activations induce a mismatch (Gusnard and Raichle, 2001; Raichle et al., 2001). […] Our results extend this knowledge by highlighting that external task engagement is accompanied by the convergence between metabolic factors and functional brain network reconfigurations.”

3) Given general issues in the field with interpreting directed connectivity (e.g., Ramsey et al., 2010; Smith et al., 2011), validation with DCM seems like an insufficient standard. The authors should be explicit about the pitfalls of both approaches (DCM and MCM) and whether they cover each other's pitfalls. For example, DCM is probably sensitive to missing nodes, while MCM won't have that issue (as far as we can tell), so the two approaches nicely complement each other in this respect. Are there pitfalls that apply to both DCM and MCM? If so, how serious are these pitfalls and can they be addressed with control analyses?

Considering that MCM and DCM use different underlying data and employ distinct models, the two approaches indeed complement each other. As such, we did not attempt to use DCM as a validation tool, but rather to assess if a comparable network structure was obtained by two independent, yet complementary approaches. As suggested by the reviewers we have addressed this by adding the following to the Discussion:

“Estimating measures of directed functional connectivity is a considerable challenge, particularly when those measures depend purely upon conditional likelihoods in the data (Smith et al., 2011), or in networks with closed cycles (Ramsey et al., 2010). […] By using a generative model and an observation (HRF) equation, DCM also avoids reliance upon making direct inference on conditional dependences in data, a challenge for many measures of directed functional connectivity (Smith et al., 2011).”

4) Given the large amount of spatial smoothing that was performed across all modalities (8 mm), how informative is a spatial correlation calculated within ROIs for MCM? One way to test for this more robustly is to perform permutation testing by comparing a BOLD connectivity map from one subject to a glucose metabolism map of a different subject (and build a null distribution through many such permutations), rather than performing a basic t-test against zero (and similarly for the comparison against rest). This will determine whether spatial similarity is truly indicative of connectivity-metabolic coupling, or whether it is observed as a function of the large smoothing kernel (and relatively small ROIs).

It is in principle possible that spatial correlations induced by smoothing may have biased MCM estimates. We directly tested this hypothesis by assessing if an original correlation of zero was artificially inflated by different levels of smoothing. Compared to the suggested permutation approach, the adopted test unequivocally addresses the potential problem. We have added the following to demonstrate that smoothing does not significantly impact the MCM findings:

Materials and methods:

“To assess the influence of spatial smoothing on MCM estimates, a control analysis was carried out testing whether a spatial correlation of zero is artificially inflated by smoothing. […] To generate a null distribution representing the influence of smoothing in these finite sampled data, this procedure was repeated 500 times for each of the 22 subjects.”

Results:

“A control analysis was performed to assess the impact of spatial smoothing on the MCM estimates (Figure 3—figure supplement 3). […] This suggests that smoothing does not systematically inflate the MCM estimates, which is also in line with our observation that half of the MCM values were approximately zero despite using smoothed data (Figure 3).”

5) Why were DCM ROIs different from the MCM ROIs (Figure 4—figure supplement 1)? If DCM is used for validation, it seems to make sense to use the same regions in both analyses. Furthermore, how does the interaction between the size of the ROIs (5 mm) and the smoothing kernel (8mm) affect the results?

As suggested by the reviewers we have re-calculated the DCM with a ROIs size of 8 mm (i.e., matching the smoothing kernel), yielding similar results. We also added an explanation about the different ROI definition for the two approaches.

BOLD methods:

“[…] the data were normalized to MNI-space and smoothed with an 8 mm Gaussian kernel to match the kernel used for the PET and ASL data.”

DCM methods:

“The BOLD signal was extracted as the eigenvariate within a sphere of 5 mm radius, adjusted for the effect of interest. […] Thus, the region employed for DCM represents a subregion of that used for MCM, with a focus on the individual location of task-specific changes in the BOLD signal.”

DCM results:

“Notably, the DCM results did not change when using a larger ROI size of 8 mm (Figure 4—figure supplement 3).”

Results:

“To allow for a focus on cross-modal changes, the regions included in MCM and DCM analyses were Occ, FEF and IPS as obtained from the conjunction analysis (Figure 2). […] The average distance between any voxel of one region to the closest voxel of another region was 46 ± 23 mm, mitigating potential cross-talk between the regions introduced through spatial smoothing.”

6) Some of the statistical tests are unclear. The authors wrote that "the significance of each connection was tested separately using one-sample t-tests against zero". So let's consider one connection: FEF to IPS. The MCM analysis yields one spatial correlation for each subject. So can I confirm the t-test involves 22 numbers (corresponding to the 22 subjects)? Because a p value of 10e-12 is quite impressive with only 22 subjects; one would need a t-score of around 14?

Correct, the t-tests of the MCM against zero comprised 22 numbers, one for each subject. The t-value of the connection FEF to IPS (hard condition) was indeed t = 14.4 and the corresponding p-value was p_uncorrected_ = 2*10^-12 and p_Bonf-Holm_ = 4*10^-11.

To address the comment, we have also provided the data used to generate Figure 3, enabling the reader to assess the validity of the statistical tests. We have also amended the manuscript as follows:

“For the MCM analysis, the significance of each connection was tested separately using one-sample t-tests against zero (6 connections * 3 conditions). As MCM yields one spatial correlation value for each subject, the test for each connection and condition is pooled across 22 correlations, corresponding to n = 22 subjects.”

7) For Figure 3, the authors say that the thickness of arrows depend on r values. More details will be useful. For example, when looking at Occ -> FEF, the two box plots for easy and hard conditions look highly overlapping, but the green arrow was much thicker in the hard condition than easy condition. And from what I gather from the plot, the IPS->FEF during hard condition has higher MCM correlation than Occ->FEF during hard condition. So why is the green arrow thickness than the yellow arrow during the hard condition?

We have now revised the figure as suggested. Furthermore, r-values were changed to z-scores. We also changed boxplots to dot plots with lines connecting the individual subjects to better reflect the within-subject nature of the design and the statistics.

8) It would be important to report the pure functional connectivity among the 3 regions. More specifically, are there region pairs with strong functional connectivity, but the MCM connectivity is weak in both directions. If so, how can this be interpreted? Similarly, are there regions with weak functional connectivity but the MCM connectivity is strong? If so, how can this be interpreted?

We have now also computed the pure functional connectivity between the three regions. The results are now included in the manuscript as follows.

Materials and methods:

“For MCM, functional connectivity was computed as seed-to-voxel correlations between two brain regions followed by z-transformation. […] For direct comparison between MCM and functional connectivity, we also report the functional connectivity as the Pearson’s correlation between time courses of the three brain regions.”

Results:

“Notably, functional connectivity was also sensitive to task performance with significant increases from rest to task for all three connections (Figure 3—figure supplement 1). Taking into account the underlying metabolic demands using MCM allowed us to infer directionality on these task-induced changes in functional connectivity, i.e., the influence one region exerts on another (Occ -> IPS, Occ -> FEF), as well as a separation of task effects (FEF -> IPS) from intrinsic connectivity (IPS -> FEF).”

9) The DCM modeling needs to be further elaborated. Maybe the current details are sufficient for a practiced DCM user, but as someone who is somewhat familiar with DCM (but do not actively use it), I found it hard to know how task modulation and intrinsic connection were modeled in the generative model.

As requested, we have improved the description of the DCM approach:

“Dynamic causal modelling (DCM): The directed functional connectivity model obtained from the MCM analysis was benchmarked against directed effective connectivity inferred from DCM. […] Each effective connection was also individually tested for the presence of task modulation using model family inference (i.e., all models comprising task modulation of a given connection against those without).”

10) "A putative mediation effect of FEF was suggested by the observation that Occ -> IPS MCM values decreased in the easy condition (from 0.17 to 0.14, pBonf-Holm = 0.017) when controlling for the indirect pathway (Occ -> FEF -> IPS)." – Please elaborate on how this is done.

We apologize for the missing information and have added the following details to the Materials and methods section:

“We tested for a potential mediation effect of FEF on the connection Occ -> IPS by adapting an established statistical approach (Baron and Kenny, 1986) to the MCM framework. […] The resulting MCM values were then compared to the original MCM values of Occ -> IPS using a paired t-test.”

11) Literature review and discussion.A) Given the nature of the paper, the authors should include a discussion of how their findings compare to a large and long-standing literature on cerebral blood flow and metabolism, which explicitly contrasts cerebral blood flow, oxygen consumption, and glucose metabolism to one another in and outside of tasks contexts, e.g., from Marc Raichle and others (see for example: Gusnard and Raichle., 2001). The current work would have substantially greater impact if it were to engage more deeply with this older literature. For example, do the authors mean to suggest (as might appear from their statement in the Abstract that "dynamic regulation of neural-metabolic coupling is essential to support human cognition") that different mechanisms exist to couple neural activity with glucose consumption depending on the task at hand? This claim seems quite large and far reaching given the evidence, and if true, would have profound implications in invalidating the current use and interpretation of many neuroimaging methods.

We agree that the paper could have been more clearly contextualized and now include a more comprehensive discussion of the physiological mechanisms underpinning CBF, oxygen consumption, and glucose metabolism (see also, our revisions in response to main issue 2, above). Notably, we did not mean to imply that the physiological mechanisms coupling neural activity to glucose metabolism (and BOLD) depend on the task at hand. As rightly pointed out, this claim goes well beyond what we can rightly infer from our data. Rather, the time-dependent, multi-step and nonlinear nature of these relationships become more apparent in the setting of a dynamic task of increasing complexity, as does the role of varying signal and noise on the statistical effects evident across different imaging modalities.

We revised the Abstract:

“The ability to solve cognitive tasks depends upon adaptive changes in the organization of whole-brain functional networks. […] Our findings show how the upregulation of key computational mechanisms to support cognitive performance unveils the complex, interdependent changes in neural metabolism and neuro-vascular responses.”

… and the impact statement:

“Cognitive performance is supported by symbiotic metabolic and neuro-vascular responses in task-specific brain networks.”

We have significantly extended the Discussion on the underlying physiological mechanisms of BOLD and [^18^F]FDG PET imaging and included the following text (see also our response to main issue 2):

“The relationship between blood flow and metabolic factors has been demonstrated to be in balance at rest, whereas task-induced neural activations induce a mismatch (Gusnard and Raichle, 2001; Raichle et al., 2001). […] Our results extend this knowledge by highlighting that external task engagement is accompanied by the convergence between metabolic factors and functional brain network reconfigurations.”

B) The authors should also be more precise about differences with previous work (e.g., Jamadar et al., 2019; Rischka et al., 2018). For example, "Here we extend upon previous work by providing a comprehensive assessment of the neural and metabolic processes supporting performance in a complex cognitive visuo-motor task" can be briefly expanded to mention that Rischka investigated a finger-tapping task and Jamadar investigated a visual checkerboard task.

As requested, we have added this information:

“To date, only a few studies using [^18^F]FDG fPET imaging have assessed the possible functional relationships between these diverse parameters of brain activity (Horwitz et al., 1995). […]. Moreover, our results directly support previous findings suggesting that regional neural and metabolic changes converge in spatially circumscribed and functionally specialized brain regions (Jamadar et al., 2019; Riedl et al., 2014; Rischka et al., 2018).”

12) The Discussion states: "Our results extend this knowledge by highlighting that the convergence between metabolic factors and functional network reconfiguration is pivotal to support cognition". However, the relationship with cognition or performance was not tested here (e.g. MCM was not shown to be stronger for those subjects who performed better). As such, I would recommend adjusting this interpretation.

This statement has been amended as follows: “Our results extend this knowledge by highlighting that external task engagement is accompanied by the convergence between metabolic factors and functional brain network reconfigurations.”